# A Longitudinal and Comparative Content Analysis of Instagram Fitness Posts

**DOI:** 10.3390/ijerph19116845

**Published:** 2022-06-03

**Authors:** Jacqueline Ahrens, Fiona Brennan, Sarah Eaglesham, Audrey Buelo, Yvonne Laird, Jillian Manner, Emily Newman, Helen Sharpe

**Affiliations:** 1Department of Clinical and Health Psychology, School of Health in Social Science, University of Edinburgh, Edinburgh EH8 9YL, UK; jacqueline.ahrens@googlemail.com (J.A.); fionabren17@gmail.com (F.B.); emily.newman@ed.ac.uk (E.N.); helen.sharpe@ed.ac.uk (H.S.); 2Scottish Collaboration for Public Health Research and Policy, School of Health in Social Science, University of Edinburgh, Edinburgh EH8 9YL, UK; audrey.buelo@phs.scot (A.B.); jillian.manner@ed.ac.uk (J.M.); 3Prevention Research Collaboration, School of Public Health, Faculty of Medicine and Health, The University of Sydney, Sydney 2006, Australia; yvonne.laird@sydney.edu.au; 4Charles Perkins Centre, The University of Sydney, Sydney 2006, Australia

**Keywords:** social media, fitness, body image, content analysis

## Abstract

Body dissatisfaction is among the most common mental health challenges experienced by women and has been identified as a risk factor for disordered eating. Research has found that exposure to social media images depicting thin, muscular bodies, often dubbed ‘fitspiration’, may contribute to body dissatisfaction. Image-centred social media platforms, such as Instagram, have rising popularity among adolescents and young adults. However, little is known about the content of images produced by different fitness-related sources, such as those from fitness brands compared with individual users, and how fitness content on social media is evolving over time. This study sought to determine whether Instagram content varied between female fitness influencers and brands and how this content changed between 2019 and 2021. A longitudinal content analysis was conducted on a sample of 400 Instagram images using a coding scheme developed specifically for this project. The scheme coded images for fit ideal body depiction, fitness focus, objectification, and sexualisation. Chi-square tests indicated that female fitness influencer content was more sexualised and portrayed more of the fit ideal, while fitness brands produced more Instagram content with a fitness focus. There were no significant overall longitudinal changes for any of the four key variables. However, when looking at longitudinal changes by account type, fitness-focused influencer content increased while fitness-focused brand content decreased over time. These findings highlight discernible differences in content produced by different Instagram account types. It points future research towards the consideration of potential moderating factors, such as account type, when exploring the impact of social media images on body image and mental health.

## 1. Introduction

Body image, a multidimensional construct that comprises cognitive, behavioural, and affective components, has been identified as a core aspect of women’s psychological and physical health [1]. Body dissatisfaction is a form of body image disturbance arising when an individual makes a negative, subjective evaluation of their physical appearance [2]. A substantial body of literature has documented that body dissatisfaction is widespread among women and is often linked with sociocultural factors such as culturally informed beauty ideals and media narratives [3,4]. Body dissatisfaction, in turn, has been established as a risk factor for various negative health consequences, such as depression, low self-esteem, and suicidal ideation [5,6,7], as well as for appearance changing behaviours, such as compulsive exercise, steroid use, and cosmetic surgery [8]. Notably, body dissatisfaction has been found to be an important predictor of disordered eating, the onset of which often occurs during adolescence [9,10,11]. To help prevent such outcomes, factors contributing to the development of body image concerns and body dissatisfaction must, therefore, be investigated.

A prominent line of research argues that body dissatisfaction is partly due to sociocultural factors, most importantly, pressures to match appearance ideals [12,13]. Mass media plays an important role in defining these appearance ideals by representing specific body types more often than others and by depicting them as more attractive and desirable [14]. In Western societies, the female body ideal has long been characterised by an extremely thin figure, while people with overweight and obesity are typically stigmatised by the media [15]. Notably, in recent years, the female body ideal has shifted from a predominantly thin to a dual-dimensional thin and toned figure [16]. The pressure to be toned as well as thin arguably makes this ideal even less attainable.

Social media have become increasingly pervasive over the past decade and have been associated with heightened body image concerns among young women [17,18]. Instagram is an image-based social media platform. In light of this visual focus, it has been proposed that Instagram may be more harmful to women’s body image than other forms of social media, with photos and videos often being carefully selected and edited in order to conform to dominant body ideals (notably, in westernised cultures, a thin, toned physique) [19]. The platform’s interactive format—which distinguishes it from traditional media—warrants attention, as the custom of liking and commenting on posts likely fosters a culture of appearance evaluation and facilitates social comparisons [20]. For many users, Instagram is also highly accessible; once installed on a smartphone, it is available anywhere, at any time, providing limitless opportunities for engagement with idealised images.

Fitness content on Instagram (also referred to as fitspiration) has become instrumental in formulating a visual representation of health today. Recent content analyses of fitspiration posts observed that the vast majority of images depicted women with thin, visibly toned bodies—known as the ‘fit ideal’—implying that only a certain body type can be fit and strengthening the association between thinness and optimal health, which has proven detrimental to women’s psychological well-being [1,21,22].

To date, the majority of social media content analyses have focused on specific hashtags (e.g., #fitspiration, #fitspo, #thinspiration) [21,23,24], and to the authors’ knowledge, no analysis has examined Instagram fitness content more generally according to the type of content creator or conducted a comparison between content from different time points. A more nuanced understanding of media messaging around weight, body shape, and body transformation—as well as any changes in this messaging across time—could shed light on key risk factors for body dissatisfaction and associated affective and behavioural outcomes.

Instagram has evolved to attract a variety of content creators, most prominently influencers and brands. Influencers are micro-celebrities who have accumulated a very large number of followers and generate profits through corporate sponsorships or by advertising their own brands [25]. Fitness influencers typically promote ideas of (self-)transformation, encouraging their followers to ‘improve’ themselves by adopting an investment-oriented attitude towards the body [26]. Fitness is thus commodified, with messaging suggesting that bodies can be remade and reshaped through consumption. Similarly, brands use Instagram to communicate with the public and to promote and sell their products [27,28]. It is worth noting that modes of commodification may vary between these two types of content creators. While fitness influencers often display their own bodies as guarantees of the quality of a service or product, capitalising on the personal connection they have established with their followers through years of content sharing [29], commercial sports brands are likely to present a wider range of people, many of whom may be professional models not known to viewers. Thus, it is perhaps surprising that research on body image and social media has not yet explored potential differences in such commercial content. Importantly, these different types of content may have different psychological effects on Instagram users; influencers tend to foster a peer-like relationship with their followers, and comparisons with peers on social media have been found to have a stronger effect on body image concerns than comparisons with celebrities or models [30,31]. Understanding differences in content produced will provide an important first step in identifying potential targets for intervention to prevent any negative effects.

Meanwhile, the COVID-19 pandemic has precipitated changes in patterns of social media usage, which may also have important implications for body image. Limits placed on physical contact have increased the use of platforms such as Instagram, which have, in turn, seen a rise in exercise and weight-focused content [32]. Due to the closure of gyms and other fitness spaces, many influencers have begun motivating their followers to stay active at home, livestreaming at-home workouts and providing at-home diet plans [33]. For fitness influencers, the incentive to produce regular and appropriate content is twofold: to maintain public engagement and to attract and secure brand promotion [28,34]. From the viewer’s perspective, more frequent exposure to such exercise and weight-related material may serve to intensify weight and shape concerns. Indeed, although research into the effects of COVID-19 is in its infancy, many individuals with eating disorders have identified the heightened media emphasis on physical activity as posing a significant challenge to their psychological well-being [33,35]. An examination of potential shifts in fitness trends on Instagram may, therefore, improve the understanding of pandemic-related risk factors for body dissatisfaction and associated disorders.

### 1.1. Theoretical Framework

Objectification theory [36] provides a theoretical framework for this study, helping to elucidate the relationship between social media usage and body image. The theory posits that girls and women are socialised to view their physical selves primarily from an observer’s perspective—that is, to self-objectify. An image is deemed objectifying when it reduces the body to an object to be evaluated by others [37]. In such images, the appearance of the body is valued above the individual’s competence or ability. In other words, the body comes to symbolise the whole person. A prevalent type of objectification is sexualisation, which occurs when women are reduced to their sexual appeal and attractiveness [38]. Although objectification can include sexualisation, they are generally considered distinct constructs, as a body can be objectified without being sexualised [20].

Viewing images of idealised and sexualised female bodies can lead to increased body checking and self-surveillance, thereby exposing the individual to bodily shame and anxiety and potentially reducing their awareness of internal bodily states [36]. This form-over-function mindset places the viewer’s objectifying perspective of the physical body above the function it serves for the self, which appears to follow a lifespan model, rising in adolescence and dissipating in midlife [39]. It is plausible that Instagram facilitates self-objectification, as users are constantly exposed to highly curated images of both peers and celebrities [40]. Interestingly, Instagram usage also appears to mirror the trajectory of objectification theory, with popularity peaking in the 18–29 age group and declining in midlife [40,41]. The opportunity to closely observe peers—rather than purportedly unattainable celebrity images—also facilitates higher levels of social comparison [42,43].

### 1.2. The Current Study

This study aimed to identify and describe key characteristics of fitness content on Instagram and establish whether these varied depending on the type of content creator (i.e., fitness influencer or brand). It also aimed to determine whether trends across this content have changed from 2019 to 2021. A clearer understanding of these aspects of Instagram fitness content could inform intervention targets, facilitating the provision of more specific support for those who experience body image issues during or following Instagram use. Due to the previously identified gaps in the literature, it was not possible to form directional hypotheses. The paper instead investigates the following exploratory research questions:

RQ1: To what extent do rates of fit ideal depiction, fitness focus, objectification, and sexualisation differ between influencer and brand Instagram fitness content?

RQ2: To what extent have rates of fit ideal depiction, fitness focus, objectification, and sexualisation in Instagram fitness content changed between 2019 and 2021?

RQ3: Are there any temporal changes in fit ideal depiction, fitness focus, objectification, and/or sexualisation that vary between influencers and brands?

## 2. Methods

### 2.1. Study Design

Ethics approval for this study was granted by the Health in Social Science Research Ethics Committee at the University of Edinburgh, Scotland. A longitudinal content analysis of fitness content shared on Instagram was conducted. A 2019 dataset was drawn from a research project by the Scottish Collaboration for Public Health Research and Policy [44], where images were collected from the Instagram feeds of 10 popular female fitness influencers and 10 popular fitness brands, previously identified through a Google search. To acquire a 2021 sample, the 2019 data collection method was replicated. The list of female fitness influencers and fitness brands was updated to reflect the popularity levels of 2021. Fitness influencers were defined by four main criteria: (1) popularity (operationalised by the number of followers) [44], (2) fitness-related content (e.g., account biography or name referring to fitness [29,44], (3) having at least one commercial partnership with a company [29], and (4) qualifying as a public figure (“blue tick” verification). Fitness brands were defined by similar criteria: (1) popularity (number of followers), (2) promoting fitness-related products, and (3) having a public account (“blue tick” verification). To identify the relevant accounts, the prompts “most followed female fitness influencers on Instagram” and “most followed fitness brands on Instagram” were entered separately into the search engine Google UK. This purposive sampling method is consistent with previous research investigating social media content [3,45]. Taken from the first page of Google search results, website articles listing top fitness Instagram accounts were screened following the aforementioned inclusion criteria to identify the relevant fitness accounts and lists of the top 10 female fitness influencers, and top 10 fitness brands of 2021 were compiled (the full list is provided in Appendix A).

### 2.2. Data Collection

The data collection was conducted on the 17 April 2019 and the 17 April 2021. Ten posts were sampled from each of the predetermined accounts, working backwards from the collection date. The researchers logged into Instagram through their personal accounts and took screenshots of the images or video thumbnails and of the associated captions. Each post was recorded by two researchers and cross-checked to ensure accuracy. The final dataset consisted of 400 images for coding and was posted between the 24 October 2018 and the 17 April 2019 and the 22 October 2020 and the 17 April 2021.

### 2.3. Coding Procedures

A coding scheme was developed, drawing on previous social media content analyses. The template for this coding scheme was a coding guide by Ghaznavi and Taylor [46] developed to analyse images of ‘thinspiration’, an Internet trend that promotes thinness and weight loss. Adaptations to the coding guide were made to more accurately capture the fitness dimension of this study and to reduce subjectivity in coding.

To ensure that features of the fit ideal body (including associated lack of fitness focus) and aspects of objectification (including sexual objectification) could be captured, variables relating to image purpose, body depiction, and sexualisation were developed based on research by Carrotte et al. [21], Deighton-Smith and Bell [22], and Tiggemann and Zaccardo [23].

To test the initial coding scheme and assess intercoder reliability, pilot coding was conducted on the first 30 images from the 2019 database. After the first pilot round, discrepancies were identified and discussed by all three researchers, and the coding scheme was further refined. Superfluous variables (e.g., ‘textual’, ‘social endorsement’) were removed. To increase objectivity, broader categories (e.g., ‘sexualised pose’, ‘body parts in focus’) were broken down into component variables (e.g., ‘back arched’, ‘wide stance’, ‘arms in focus’). Drop-down lists of codes were created for each variable in the Excel coding template to ensure consistent use of terminology. A second pilot round was then conducted on the same sample, and intercoder reliability was calculated for each pair of coders using Cohen’s kappa. The resulting kappa value indicated substantial agreement, κ pilot = 0.83 [47], validating the use of the developed coding scheme. Following this coding scheme refinement, three researchers went on to code all 400 images. Images were divided between pairs to be coded independently: pair 1 (*n* = 160), pair 2 (*n* = 120), and pair 3 (*n* = 120). A final codebook with instructions and examples was used to guide coding (see Appendix A). The coders aimed to capture the main messages and clear implications of the post, avoiding subjective judgments. Captions and hashtags could be used to give context to the post. In the case of a photo series, only the lead photo was coded. In the case of video content, the thumbnail was coded. If an image depicted both a woman and a man, the woman was coded. If a group of women was depicted, the dominant figure—identified through examination of the image structure, in line with guidance from Carrotte et al. [21]—was coded.

### 2.4. Coding Attributes

#### 2.4.1. Image Type and Content

These covered basic features of the image: its format (still image, photo series, video content); the category of subject matter (people or other); demographic characteristics of any human subjects depicted (gender, age, ethnicity); and information regarding location (fitness space, residential property, indoor other, outdoor other, unclear, and whether exercise equipment was present or not present).

#### 2.4.2. Image Purpose

Based on a scoping of the relevant literature, three common types of image purpose were identified: promotion, education, and motivation. Each of these categories was further broken down to capture different underlying aims: the promotional, educational, or motivational content of a post could be fitness-related (i.e., providing workout guidance, referencing physical fitness or fitness apparel/equipment), appearance-related, or neither of these (i.e., ‘other’).

For the purpose of the present analysis, it was important to distinguish promotional posts from educational or motivational posts, as those containing promotion (particularly where this involved a body or body part) were deemed more likely to contain elements of objectification. In light of recent findings regarding the potentially negative effects of appearance-focused exercise [48], it was also important to attempt to differentiate fitness-focused posts from those that either contained no meaningful fitness-related information or overtly emphasised appearance concerns.

#### 2.4.3. Body Depiction

The coding of body depiction drew on recent fitspiration literature and objectification theory. Variables coding for thinness and muscularity were added as they have repeatedly been identified as important characteristics of sociocultural appearance ideals [49]. The subject’s level of thinness (low body fat or not) and muscularity (little to no definition, visible definition, and high-level definition) was coded in line with previous research by Deighton-Smith and Bell [22] and Tiggemann and Zaccardo [23]. Further, variables were modified to more directly capture objectification and sexualisation. Since Fredrickson and Roberts [36] identify the elimination of a person’s head or face from an image as a form of objectification, face visibility was coded. In line with the concept of body fragmentation, images were coded for whether they emphasised specific body parts. Following Carrotte et al. [21], body parts emphasised were identified by examining visual cues such as proximity to the camera and cropping. Specific decisions as to which body parts to code (namely, glutes, abs, arms, chest, and legs) were based on the researchers’ initial observations of the dataset. The proportion of the body shown in the image was also coded according to an amended version of the scale established by Deighton-Smith and Bell (namely, 100%, 75%, 50%, or 25% body visibility) [22].

Following a scoping of the relevant literature and the 2019 data, five common features of sexualised poses were identified: glutes pushed towards the camera, back arched, wide stance, pulling at clothing/hair, and one foot forward to emphasise glutes. These were listed as separate variables as more than one could be present in an image. The following features of sexualised facial expressions were also coded: pouting, direct gaze, mouth open, and biting lip/tongue. Finally, in keeping with prior research by Deighton-Smith and Bell [22], type of clothing was coded (activewear, sexualised clothing, everyday clothing), as well as fit (tight-fitting, standard fit, swimsuit/underwear) and amount of skin exposed (i.e., extremely revealing, very revealing, moderately revealing, not at all revealing).

### 2.5. Intercoder Reliability

Once coding was completed, intercoder reliability was calculated for each pair of coders to ensure that the coding scheme had been interpreted consistently. The mean Cohen’s kappa indicated high reliability (κ = 0.85), and the mean kappa values for each variable (other than ethnicity and age, for which coding was based on internet searches where possible) ranged between 0.63 and 0.95, indicating moderate to near-perfect agreement [47]. One variable (‘Motivational’) had a particularly low average kappa value of 0.47 (see Appendix A for all kappa values). However, since this variable was not used directly in the analysis, it was not recoded. To compose the final dataset, the remaining discrepancies were resolved collaboratively via discussion between the three coders.

### 2.6. Data Analysis

To answer the set research questions, codes were transformed into numerical data, and four binary variables were computed: fit ideal body depiction (yes/no); fitness focus (yes/no); objectified image (yes/no); and sexualised image (yes/no). An image was coded as presenting the fit ideal body if it depicted a person with low body fat and either visible or high-level muscular definition. An image was coded as fitness-focused if it met two or more of the following criteria: depicts fitness equipment or set within a fitness space (or both); conveys an educational, promotional, and/or motivational message that is fitness-focused; depicts a person actively engaging in exercise; depicts a person wearing activewear. An image was coded as objectified if it met two or more of the following criteria: face not visible, at least one body part in focus, small proportion of body visible (25–50%). Finally, an image was coded as sexualised if it met two or more of the following criteria: sexualised facial expression (based on having two or more markers); sexualised clothing (based on having two or more markers); sexualised pose (based on having two or more markers). While there is currently no set standard for defining these constructs within social media research, the threshold of two criteria was intended to minimise the likelihood of false positives.

To address RQ1 and RQ2, chi-square tests of association were performed to test for statistically significant differences between influencer content and brand content and images from 2019 and those from 2021. Phi coefficients were used as a measure for effect size. To address RQ3, binary logistic regression analyses were used to evaluate the impact of account type and year on the prevalence of our variables of interest: fit ideal, fitness focus, sexualisation, and objectification.

Where the variable under analysis related specifically to body depiction (e.g., sexualisation, fit ideal body depiction), only images of people were included in the analysis. In other cases (e.g., fitness focus), all images were analysed. Since multiple comparisons were computed, adjustments for multiple testing were made to capture true effects and to minimise type 1 error inflation. A Bonferroni correction was conducted: the *p* value of 0.05 was divided by the average number of tests run for each variable (i.e., three tests). Thus, effects were designated as significant at a *p* < 0.017 threshold level. Analyses were conducted using IBM SPSS statistical software version 25 [50].

## 3. Results

### 3.1. Sample Characteristics

Of the 400 posts analysed (*n* = 200 per account type), most images were of people: 182 influencer images (91%) and 166 brand images (83%). Across images depicting people (*n* = 348), most were female (77.3%), white (47.1%), and aged between 21–35 (83.3%). (Frequencies for all descriptive variables can be found in Appendix A).

### 3.2. RQ1: To What Extent do Rates of Fit Ideal Depiction, Fitness Focus, Objectification, and Sexualisation Differ between Influencer Content and Brand Content?

Across images of people (*n* = 348), 169 images (48.6%) depicted the fit ideal body, 59 (17%) were coded as objectified, and 65 (18.7%) were coded as sexualised. Of all images (*n* = 400), 187 (46.8%) were rated as being fitness focused.

There was a statistically significant association between account type and fit ideal body depiction, χ^2^ (1, *n* = 348) = 55.25, *p* < 0.001, *phi* = −0.40, fitness focus, χ^2^ (1, *n* = 400) = 30.38, *p* < 0.001, *phi* = 0.28, and sexualisation, χ^2^ (1, *n* = 348) = 68.28, *p* < *0*.001, *phi* = −0.44. Compared with influencers, brands were less likely to depict the fit ideal body, more likely to post fitness-focused content, and less likely to post sexualised images (see Table 1). Using Cohen’s [24] criteria, the phi coefficient value indicated small to medium effect sizes across these associations. In contrast, there was no significant association between account type and objectification, χ^2^ (1, *n* = 348) = 5.43, *p* = 0.020, *phi* = −0.13, meaning that rates of objectification did not differ between images posted by influencers and brands.

### 3.3. RQ2: To What Extent Have Rates of Fit Ideal Depiction, Fitness Focus, Objectification, and Sexualisation in Instagram Fitness Content Changed between 2019 and 2021?

There was no statistically significant association between year and any of the content variables, i.e., fit ideal body depiction, fitness focus, objectification, or sexualisation χ^2^ (1, *n* = 348) 5.44, *p* ≥ 0.020, *phi* ≤ 0.13. This demonstrated that, overall, the content of images in these domains did not change between 2019 and 2021 (see Table 1).

### 3.4. RQ3: Are There Any Temporal Changes in Fit Ideal Depiction, Fitness Focus, Objectification, and/or Sexualisation That Vary between Influencers and Brands?

There was no significant interaction between account type and year for fit ideal body depiction (β = −0.65, S.E. = 0.48, *p* = 0.175), objectification (β = −0.57, S.E. = 0.62, *p* = 0.353), or sexualisation (β = −16.98, S.E. = 4493.71, *p* = 0.997; see Table 2), meaning that the temporal change was similar between influencers and brands in these domains (see Appendix A for descriptive statistics related to these models).

In contrast, there was a significant interaction between account type and year for fitness focus in content (β = −1.46, S.E. = 0.43, *p* = 0.001; see Table 2). The rate of fitness focus in influencer content rose over time, with influencers being more likely to post a fitness-focused image in 2021 compared with 2019 [χ^2^ (1, *n* = 200) = 5.79, *p* = 0.016]. Conversely, the rate of fitness focus in brand content dropped slightly, with brands being less likely to post a fitness-focused image in 2021 compared with 2019 [χ^2^ (1, *n* = 200) = 6.05, *p* = 0.014); see Figure 1]. Across the two time points, the overall rate of fitness focus remained higher in content posted by brands.

## 4. Discussion

### 4.1. This Work

The present study addressed two notable gaps in the literature. To the researchers’ knowledge, the study was the first to compare different types of content creators and, therefore, provided insight into the range of material posted on the platform, paving the way for greater awareness of the potential effects of consuming influencer or brand content. Relative to brands, fitness influencer content had a greater depiction of the fit ideal, fitness focus, and sexualisation, while there were no differences in objectification. Influencer posts, in particular, were, therefore, found to contain several features that the existing research has identified as having potentially deleterious effects on women’s body image [19,51,52]. In addition, the present study is the first to perform a temporal comparison of Instagram fitness content, thus contributing to the current understanding of the development of fitness content over time. Overall, most of the content remained stable over time. The only exception was rates of fitness-focused content: brand content became less fitness-focused, while fitness influencers increased their production of fitness-focused content over time.

The finding that fitness content varied between influencers and brands is noteworthy, as this was the first study to examine potential differences between different types of content creators. In light of the possibility that influencer content could have a stronger effect on body image concerns than brand content (due to the peer-like nature of influencers’ relationships with their followers) [30], the prevalence of the fit ideal in influencer content is particularly significant. In line with findings from previous content analyses, fitness influencers frequently depicted the fit ideal, with over two-thirds of images showing women who were both muscular and thin [23,45,46]. Conversely, a little over a quarter of fitness brand content promoted this image. Since internalisation of the fit ideal has been linked to increased body dissatisfaction among women—likely because the pressure to be toned as well as thin makes this ideal even less attainable—exposure to the kind of influencer fitness content examined here may have negative consequences for women’s body image [53].

The higher proportion of sexualised images in influencer content should also be noted. Female fitness influencers were found to frequently depict the fit ideal and sexualisation while sharing less fitness-focused content than fitness brands. Therefore, it appears that female fitness influencers use their visibility to reinforce narrow norms of self-presentation on Instagram, which are likely to perpetuate unrealistic body ideals, encourage the sexualisation of women, increase self-objectification and lead to higher levels of body dissatisfaction. Furthermore, since Instagram’s algorithm models a user’s online environment based on previous likes and searches, followers of one of these fitness influencers are likely exposed to content from other similar fitness influencers, which may inadvertently increase the risk for these negative outcomes [54]. Consequently, female fitness influencers may play a relevant role in shaping women’s body image. It should be noted, however, that the rate of sexualisation in brand content was extremely low, pointing towards a level of diversity in Instagram fitness content that tends to be obscured in studies that do not distinguish between different types of accounts. One practical implication of this finding is that media literacy programmes could be adapted to include information on content shared by fitness influencers in order to reduce its potential impact on body image concerns.

Although fitness focus remained higher in brand content, fitness influencers did increase the amount of fitness-focused content shared between 2019 and 2021. As influencers are more dependent on the engagement of their followers [55], they might have been under more pressure to diversify their content to reflect the changing demands of their followers. As such, this finding aligns with Godefroy’s [29] initial observation that in response to COVID-19, fitness influencers changed their content to increasingly adopt the role of sports coaches. Brands, on the other hand, already shared more fitness-focused content before the pandemic and thus might not have had to adapt their content in the same way. Future research should monitor how this trend develops in the coming years.

Finally, the overall lack of temporal change observed warrants attention. While emerging research into media exposure during COVID-19 has pointed towards a rise in exercise and weight-related content, as well as increased demand for fitness education and heightened pressure on influencers to remain visible [32,33], the analyses here revealed no statistically significant differences between 2019 and 2021 (although rates of objectification and sexualisation did increase slightly). It is possible that methodological decisions prevented some potential temporal shifts from being identified: a relatively small number of accounts were examined, and a large number of the posts analysed were in video format (which could not be coded in full). Moreover, the two-year time frame may not have been sufficient to identify longer term shifts.

### 4.2. Limitations

A number of study limitations must be acknowledged whilst interpreting these results. First, the findings are limited by the decisions made regarding attributes and themes included in the coding scheme. Some of the variables included were subjective and thus more susceptible to erroneous reporting (notably sexualisation and objectification) [20]. Although the definition and conceptualisation of these variables will undoubtedly evolve over time, it is hoped that the rigorous experimental process implemented in the present study best captured the appropriate definition for each term in today’s society. It is also important to recognise that the coding scheme was developed for use on still images, meaning that little detail could be provided on video content (which made up a notable proportion of the dataset). This analysis is, therefore, limited to static images and video thumbnails. Furthermore, it should be noted that engagement in the form of endorsement (e.g., the number of likes) and comments posted with the images was not recorded in this study. Such data may have helped to gauge the relationship between content features and the level of interaction achieved by the post but are also more challenging to capture on social media as older images exhibit greater variance in the amount of engagement than more recent ones. Given that Instagram users post with varying degrees of frequency, it should also be noted that the selection of ten posts for each Instagram account represents a different time frame relative to the point of data collection (17 April). Finally, the study does not include data for 2020 and is, therefore, unable to see how Instagram content style may have been impacted at this time.

### 4.3. Future Work

Moving forward, this study could be extended in a number of key ways. It could serve as a template for further content analyses of fitness content on social media. Future research should aim to replicate these results with more detailed datasets and over a longer timeframe. Whilst we focused on popular fitness accounts, sampling content using other methods (e.g., random sampling approaches or focusing on particular ‘subcultures’) may help to provide a more nuanced understanding of how content is changing. Since men also consume fitness content on Instagram, a similar study of male fitness influencers could be conducted to consider the potential effects of this kind of material on men’s body image. The two time points compared here (2019 and 2021) were relatively close together, so further longitudinal research may be necessary to identify longer-term temporal shifts. Since, as Perloff [31] points out, media effects are transactional, experimental research is needed to investigate whether the observed differences between influencer and brand content are reflected in the effects of this content on Instagram users, taking into account individual difference factors. The recent development of an assessment tool for women’s internalisation of the fit ideal [53] should facilitate such future research.

## 5. Conclusions

This study has demonstrated that fitness content on Instagram, especially that posted by influencers, displays several characteristics which have been identified as potentially harmful to women’s body image. These findings illustrate the complexity of social media’s influence on body image and set the stage for a number of avenues for future research. Since influencers and brands differ in terms of fit ideal depiction, fitness focus, and sexualisation—all of which have previously been linked to body image disturbance among women [52,53,56]—future research into the impact of social media usage on body image should pay greater attention to the specific source of Instagram fitness content. If potentially harmful content can be traced to specific account types on social media, content creators associated with those account types could be targeted directly by intervention efforts to increase social media literacy. The better we understand different types of Instagram content and its implications for viewers, the better we can inform intervention and prevention efforts—both to encourage social media platforms to introduce more robust safeguarding policies and to help social media users to navigate their way through these online fitness spaces.

## Figures and Tables

**Figure 1 ijerph-19-06845-f001:**
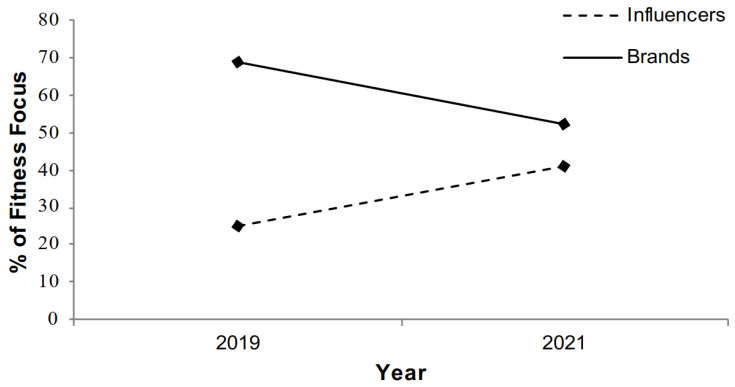
Changes in Fitness Focus by Account Type and Year.

**Table 1 ijerph-19-06845-t001:** Distribution of Fit Ideal, Fitness Focus, Sexualisation, and Objectification in Images by Account Type and Year.

	Account Type				Year			
	Influencer		Brand		2019		2021	
	*n*	%	*n*	%	*n*	%	*n*	%
Fit Ideal (*n* = 348) Yes No Total	123 59 182	67.6 32.4 100	46 120 166	27.7 72.3 100	87 85 172	50.6 49.4 100	82 94 176	46.6 53.4 100
Fitness Focus (*n* = 400) Yes No Total	66 134 200	33 67 100	121 79 200	60.5 39.5 100	94 106 200	47 53 100	93 107 200	46.5 53.5 100
Sexualisation (*n* = 348) Yes No Total	64 118 182	35.2 64.8 100	1 165 166	0.6 99.4 100	29 143 172	16.9 83.1 100	36 140 176	20.5 79.5 100
Objectification (*n* = 348) Yes No Total	39 143 182	21.4 78.6 100	20 146 166	12 88 100	21 151 172	12.2 87.8 100	38 138 176	21.6 78.4 100

**Table 2 ijerph-19-06845-t002:** Logistic Regression Predicting Likelihood of Outcome Variables.

		β	S.E.	*p*	Odds Ratio	95% C.I. for Odds Ratio	
						Lower	Upper
Fit Ideal	Year	0.01	0.32	0.969	1.01	0.54	1.89
	Account Type	−1.40	0.32	<0.001	0.25	0.13	0.46
	Year x Account type	−0.65	0.48	0.175	0.52	0.20	1.33
	Constant	0.73	0.23	0.002	2.07		
Fitness Focus	Year	0.74	0.31	0.017	2.09	1.14	3.81
	Account Type	1.90	0.32	<0.001	6.68	3.59	12.41
	Year x Account type	−1.46	0.43	0.001	0.23	0.10	0.54
	Constant	−1.10	0.23	<0.001	0.33		
Objectification	Year	0.88	0.39	0.022	2.413	1.13	5.13
	Account Type	−0.33	0.47	0.486	0.721	0.29	1.81
	Date x account type	−0.57	0.62	0.353	0.565	0.17	1.89
	Constant	−1.82	0.31	<0.001	0.162		
Sexualisation	Year	0.22	0.31	0.486	1.24	0.67	2.29
	Account Type	−3.71	1.03	<0.001	0.02	0.00	0.18
	Year x Account type	−16.98	4493.71	0.997	0.00	0.00	
	Constant	−0.73	0.23	0.002	0.48		

## Data Availability

Data available on request due to ethical restrictions.

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
