# Peer review of "A Longitudinal and Comparative Content Analysis of Instagram Fitness Posts"

_ijerph, 2022, doi:10.3390/ijerph19116845_

Round 1

Reviewer 1 Report

Clarify the rationale for the comparison between influencers and brands and how the study findings can be implemented in academia and public health. 
Line 147. Specify the country of the University of Edinburgh. 
Line 151. how are 10 popular female fitness influencers and 10 popular fitness brands are selected? Did the google search do in a specific culture or country? 
The duration between 2019 and 2021 is too short for the comparison of changes in fit ideal depiction, fitness focus, objectification, and sexualisation in Instagram fitness content. What is the rationale of these years' comparison?
Line 169. if The data collection was conducted on the 17th of April 2019 and the 17th of April 2021, what's the date of each image posted on Instagram? Although you collect data 2021, the images might not be posted and shared in 2021. 
Line 203. Elaborate rationales and definitions of each coding attribute. 
Gender is an important factor to influence body image study. Is there any gender difference in the objectification of fitness content? 

Reviewer 2 Report

Dear Authors; This is very interesting study in the age of social media. I find its conceptualization and statistical analysis solid. However, its presentation needs some "minor work" to arrive to journal standards. Regards.

P.S.

[1] Writing:

1-1 Author Affiliations: Add city, state, country

1-2 References: Check out MDPI format. For example, years for papers are in bold font.

1-3 Sections/Subsections: Add numbers to them in hierarchical way to show they are in order and nested. It is difficult to follow the paper at the present status.

1-4 Discussion: It is messy at the current status. Make it sectionalized like this: 4.Discussion, 4.1. This work, 4.2. Limitations, 4.3. Future work

1-5 Conclusion. Finish the paper with "5.Conclusion" in a 5-6 lines paragraph.

[2] Statistical:

2-1: Line 288 : SPSS software needs its own citation in the reference section:

[citation] IBM Corp. Released 2020. IBM SPSS Statistics for Windows, Version 27.0. Armonk, NY: IBM Corp

2-2: How did you match two datasets in 2019 and 2021 ? Did you use propensity scores, matching or stratification, or other method ? Add the answer in couple sentences in line 186.

Reviewer 3 Report

The title of the manuscript is interesting however, I would like to draw your attention to issues that should be addressed and resolved. Please find below my comments.
1. Introduction is more like a literature review. The motivation and significance of the research are weakly addressed in the introduction section. Please provide a strong justification for your study. It would be more desirable if needs, gaps, and objectives can be further emphasized in the Introduction section. In addition, the sub-section "The Current Study" (Lines: 129 - 144) should be merged under the introduction section.

2. The setion "Theoretical Framework" (Lines: 92- 128) should be extended by creating a new section (Literature Review). I did not find any literature review section to support this study. This suggests that the authors have overlooked the most important developments in this field.

3. The results needs to be better integrated in the existing literature on the comparative content analysis of Instagram fitness posts.

4. The discussion section is shallow. There is a strong need to beef up the discussion (e.g., theoretical contributions and practical implications) on the findings of this study. The presented implications is missing. 

Overall, I think this manuscript needs more scientific revisions before submission for publication.

Reviewer 4 Report

The research is an empirical experiment on analyzing the fitness-related contents on the Instagram social media platform by famous brands and influencers and how it has changed over time. 

The results of the experiment and discussions are backed by the datasets and statistical analysis. However, as noted by the authors, the major limitations of the study are:

  1. The samples are taken from only two years, 2019 and 2021. Therefore, the result does not give us a perspective across a longer timeframe and how the contents have shifted. It also misses the data for 2020 with Covid pandemic and how it might have impacted the changes in the content styles. 
  2. The two categories of brands vs. influencers are very broad and may not capture the variations within these groups such as different sports, age groups, locations, number of page followers, etc. and the data is biased towards most famous or top google search results. It should be noted that companies and influencers pay for advertisements and sometimes fake accounts to inflate their ratings and number of followers in social networks. 
  3. Moreover, since this study could be a complement to other researches that focus on the impact of social media content on women's body image, it would be ideal to add the number of people who liked these contents, commented, etc. and possibly an estimate of the distribution of male/female user who liked the contents to the dataset to capture the weight of influence of these contents on different genders. 

Overall, this study could be used as a template to further replicate the results with more detailed datasets and over a longer timeframe. 

Round 2

Reviewer 1 Report

Line 151. how are 10 popular female fitness influencers and 10 popular fitness brands selected? Did the google search done in a specific culture or country? The author answered that the influencers and brands were identified through a Google search. Please elaborate on this google search. What kinds of keywords were used in these searches, and what kinds of information or/and data were used. Also, google USA or UK? 

The data from Instagram were posted in 2019 (24/10/2018 – 17/04/2019) and 2021 (22/10/2020 – 17/04/2019). Then, actually, the data presented in 2021 were from 2019 to 2020 but not 2021. Also, the data includes posts before the Pandemic started. there is no rationale to indicate whether these data could be relevant for the longitudinal study. 

Author Response

Response to Reviewer 1 Comments

Line 151. how are 10 popular female fitness influencers and 10 popular fitness brands selected? Did the google search done in a specific culture or country? The author answered that the influencers and brands were identified through a Google search. Please elaborate on this google search. What kinds of keywords were used in these searches, and what kinds of information or/and data were used. Also, google USA or UK?

The Google search included the terms “most followed female fitness influencers on Instagram” and “most followed fitness brands on Instagram”. Each researcher used Google UK to search these terms before compiling a shared list of most followed Influencers and brands. These lists were ultimately used to identify the top 10 for each account type.

This can now be seen on lines 191 – 199. We have specified the use of Google UK on line 193.

The data from Instagram were posted in 2019 (24/10/2018 – 17/04/2019) and 2021 (22/10/2020 – 17/04/2019). Then, actually, the data presented in 2021 were from 2019 to 2020 but not 2021. Also, the data includes posts before the Pandemic started. there is no rationale to indicate whether these data could be relevant for the longitudinal study.

Please accept our apologies and excuse the typing mistake - the date range for the second data collection should read 22/10/2020 - 17/04/2021. The first collection of images were posted and before the pandemic began (24/10/2018-17/04/2019) and the second collection of images were all posted during the pandemic (22/10/2020 - 17/04/2021). This is in line with our rationale in which we aimed to identify possible shifts in fitness trends in Instagram content over time.

Reviewer 3 Report

The authors addressed all the given comments.

Author Response

Response to Reviewer 3 Feedback

The authors addressed all the given comments.

Thank you for accepting our responses, it is greatly appreciated.